# Effects of *Caulerpa taxifolia* on Physiological Processes and Gene Expression of *Acropora hyacinthus* during Thermal Stress

**DOI:** 10.3390/biology11121792

**Published:** 2022-12-09

**Authors:** Jian-Rong Fu, Jie Zhou, Yan-Ping Zhang, Li Liu

**Affiliations:** 1Fisheries College, Guangdong Ocean University, Zhanjiang 524088, China; 2Guangdong Laboratory of Southern Ocean Science and Engineering, Zhanjiang 524025, China

**Keywords:** *Acropora hyacinthus*, *Caulerpa taxifolia*, thermal stress, physiological processes, gene expression

## Abstract

**Simple Summary:**

This study explored the crucial issue of how the physiology and molecular response of the hermatypic coral *Acropora hyacinthus* was affected by the foliaceous macroalgae *Caulerpa taxifolia* at various temperatures. Therefore, the different ways including indirect and direct coculturing with seaweed were set up at an ambient temperature (27 °C) and at a +3 °C increase. Firstly, the results demonstrated that macroalgae could prominently trigger a drop in the density of zooxanthellae at various temperatures, which has been associated with the biological processes of vesicle transport, autophagy, and apoptosis regulated by the Rab5, ATG5, and Casp7 transcription factors. Secondly, oxidative stress (CAT, SODC, HPS family) and microbial immune response (IFI47, TRAF family) biological processes were violently aggravated by heat stress, resulting in cell apoptosis, at which point *Caulerpa taxifolia* alleviated the pressure influences.

**Abstract:**

An increasing ecological phase shift from coral-dominated reefs to macroalgae-dominated reefs as a result of anthropogenic impacts, such as eutrophication, sedimentation, and overfishing, has been observed in many reef systems around the world. Ocean warming is a universal threat to both corals and macroalgae, which may alter the outcome of competition between them. Therefore, in order to explore the effects of indirect and direct exposure to macroalgae on the physiological, biochemical, and genetic expression of corals at elevated temperature, the coral *Acropora hyacinthus* and highly invasive green algae *Caulerpa taxifolia* were chosen. Physiologically, the results exhibited that, between the control and direct contact treatments, the density and chlorophyll a content of zooxanthella decreased by 53.1% and 71.2%, respectively, when the coral indirectly contacted with the algae at an ambient temperature (27 °C). Moreover, the enzyme activities of superoxide dismutase (SOD) and catalase (CAT) in coral tissue were enhanced by interacting with algae. After an increase of 3 °C, the density and chlorophyll a content of the zooxanthella reduced by 84.4% and 93.8%, respectively, whereas the enzyme activities of SOD and CAT increased 2.3- and 3.1-fold. However, only the zooxanthellae density and pigment content decreased when *Caulerpa taxifolia* was co-cultured with *Acropora hyacinthus* at 30 °C. Molecularly, different from the control group, the differentially expressed genes (DEGs) such as Rab family, ATG family, and Casp7 genes were significantly enriched in the endocytosis, autophagy, and apoptosis pathways, regardless of whether *Acropora hyacinthus* was directly or indirectly exposed to *Caulerpa taxifolia* at 27 °C. Under thermal stress without algae interaction, the DEGs were significantly enriched in the microbial immune signal transduction pathways, such as the Toll-like receptor signaling pathway and TNF signaling pathway, while multiple cellular immunity (IFI47, TRAF family) and oxidative stress (CAT, SODC, HSP70) genes were upregulated. Inversely, compared with corals without interaction with algae at 30 °C, the DEGs of the corals that interacted with *Caulerpa taxifolia* at 30 °C were remarkably enriched in apoptosis and the NOD-like receptor signaling pathway, including the transcription factors such as the Casp family and TRAF family. In conclusion, the density and chlorophyll a content of zooxanthella maintained a fading tendency induced by the macroalgae at ambient temperatures. The oxidative stress and immune response levels of the coral was elevated at 30 °C, but the macroalgae alleviated the negative effects triggered by thermal stress.

## 1. Introduction

Coral reefs are the rainforests of marine ecosystems [1]. However, human activities have been contributing to global warming, leading to a continuous deterioration of the coral coverage in recent years [2]. A subsequent phenomenon is the coral reef’s succession to an algal bed [3]. Factually, macroalgae are critical communities that play important roles in stabilizing the reef structure, generating primary productivity in reef areas [4]. Nevertheless, the niche competition between partial macroalgae and hermatypic coral is drastic. The competition mechanisms in macroalgae are primarily through the retraction of polyps caused by physical contact and injury by pathogenic microorganisms or lipid-soluble compounds [5,6,7], resulting in reduced calcification of coral growth, fecundity, survival, and settlement rate [8,9,10,11,12]. However, the extent of the impact induced by macroalgae on corals varies between species and exposure conditions [13]. Some studies suggest that competition among macroalgae becomes more manifest in the context of coral reef degradation, perhaps because weakened corals do not have adequate energy to compete for space as they need to maintain various functions [14,15]. As a result, coral reef ecosystems face severe challenges from the invasion of macroalgae.

A transcriptome is a subset of genes expressed by an organism that induce gene transcription under specific circumstances to alter different molecular mechanisms [16,17]. However, the impact of ambient pressure on corals is generally only determined by apparent physiological damage [18]. However, the regulation of gene expression is prevalent before the injury [19]. For example, the influences of thermal stress on coral growth plasticity, bleaching tolerance, and skeletal properties might be related to zooxanthellae communities, microbiome composition, and coral gene expression patterns [20,21,22]. Hermatypic coral adapted to the rising temperature through the positive regulation of innate immune responses, protein ubiquitination, and apoptosis [23]. At present, it is universally acknowledged that transcriptome sequencing technology is beneficial for enlightening and evaluating how ecological factors affect scleractinian corals. Davies et al. [24] discovered the ascended transcription of the H^+^ transporter gene in corals under stress from ocean acidification, confirming the role of proton transport in promoting calcium production in the presence of strong *p*CO_2_.

Conspicuously, transcriptome sequencing has been used in corals, while the expression regulation of genes at various temperatures has not been reported in exploring the spatial interactions of macroalgae–coral. To investigate this, the branch-like coral *A. hyacinthus* and macroalgae *C. taxifolia* were selected as the study species. In order to evaluate the physical and chemical infections of seaweed, a direct-contact group and an indirect-contact (exposed) group were set up. Through a comprehensive analysis of the physiological manifestations combined with gene expression in *A. hyacinthus*, the results can be used to explore the potential ecological impact of macroalgae on reef-forming corals and provide basic data and references for understanding the relationship between corals and macroalgae.

## 2. Materials and Methods

### 2.1. Experimental Samples

*A. hyacinthus* and *C. taxifolia* were collected from Xuwen Coral Reef National Nature Reserve (109°55′ E, 20°16′ N) and transported to the laboratory. The coral was segmented into approximately 4 cm nubbins. Then, the nubbins were epoxied on the ceramic bases, and cultured in two 200 L tanks at a temperature of 26.5 °C, pH of 8.0, salinity of 33, and 200 µmol photons m-2s-1 with a 12:12 h light/dark cycle for 3 months.

### 2.2. Experimental Design

Prior to the experiment, 54 coral nubbins were randomly allocated into 18 aquariums (10 L). Two temperature treatments were set by increasing from the ambient temperature (27 °C, L) to stress temperature (30 °C, H) at a rate of 1 °C per day [25]. Equal amounts of macroalgae (25 g) interacted with the coral nubbins in the following three ways. (1) No algae were added to the culture system, i.e., the control group (D, Figure 1A). (2) The aquariums in which the corals were exposed to the aquaculture water that flowed through external algae box were referred to as the indirect contact group (H, Figure 1B). (3) The algae and corals were cultured in the same aquarium, in contact with each other, but at the same height. This treatment was set as the direct contact group (M, Figure 1C). Therefore, there were six treatments in this experiment, namely, the control group with ambient temperature—LD, the indirect contact group with ambient temperature—LH, the direct contact group with ambient temperature—LM, the control group with higher temperature—HD, the indirect contact group with higher temperature—HH, and the direct contact group with higher temperature—HM. Three coral nubbins were placed in each aquarium and each treatment included three aquariums. Half of the seawater in each aquarium was replaced every 3 days. The coral samples were collected after four weeks of incubation.

### 2.3. Determination of Physiological and Biochemical Indexes

#### 2.3.1. Sample Collection

After four weeks of incubation, three nubbins were collected from each group and rinsed with 4 °C sterilized seawater. The slurry was homogenized into six portions of 12 mL and centrifuged at 4000 rpm for 10 min at 4 °C. The precipitate was used to determine the density and Chl *a* content of the zooxanthellae. After drying, the skeletal surface area was determined using the aluminum foil technique [26].

#### 2.3.2. Zooxanthellae Density and Chl a Content

Three parts of the pellet were suspended in 5 mL formaldehyde to measure the zooxanthellae density using a microscope with a blood counting plate. Another portion was resuspended in 8 mL methanol. The pigments were extracted at 4 °C for 24 h. The extract was centrifuged (4000 rpm min^−1^, 4 °C, 10 min), and the Chl *a* was determined according to a UV–visible spectrophotometer [27]. The data were normalized to the skeletal surface.

#### 2.3.3. Growth Rate

A technique was used to measure the growth rate whereby the nubbins were put on the bottom of a beaker containing 1 L of filtered seawater (27 °C, salinity 32). The measurements were repeated every 7 d in mg cm^−2^d^−1^ [28].

#### 2.3.4. SOD and CAT

A total of 50 mL of homogeneous solution was centrifuged using a freezing centrifuge (4000 rpm min^−1^, 10 min, 4 °C), and the quantitative supernatant was collected to measure the SOD and CAT activities determined in the dilution using kits (A001-1-1, A007-1-1, Nanjing Jicheng, Nanjing, China), and finally a BCA kit to determine the protein concentration (A045-3-1, Nanjing Jicheng, Nanjing, China). The total protein content of each sample was determined, and the activity unit of the two enzymes was standardized as U mgprot^−1^.

### 2.4. Transcriptome Sequencing and Analysis of A. hyacinthus

#### 2.4.1. RNA-Seq Data Analysis

At the end of the experiment, three corals were quickly taken from each group, slightly shaken in PBS, and stored in liquid nitrogen for RNA extraction and transcriptome sequencing. The total RNA from each tissue was extracted using the Trizol method [29]. After detecting the quality of the RNA samples by agarose gel electrophoresis, 18 eligible samples were constructed and sequenced by Beijing Baimike Biotechnology.

The raw reads were filtered, and clean reads were obtained for splicing and assembly analysis. Secondly, Trinity (v2.5.1) was used to perform de novo assembly on the clean reads to obtain transcripts and remove redundancy. The longest transcript in each transcript cluster was selected as a single gene (unigene) for subsequent analysis. Eventually, the unigene was compared with the SwissProt, COG, KOG, GO, KEGG, Pfam, and eggNOG databases, and the E-value was <1 × 10^−5^.

All gene-matched transcripts from the 18 libraries were normalized and the differential gene expression analysis was performed using the software Deseq2. The differential groups were selected as LH vs. LD, LM vs. LD, HD vs. LD, HH vs. HD, and HM vs. HD. The DEGs screening threshold was false discovery rate (FDR) < 0.001 and log_2_ Fold Change ≥ 2. ClusterProfiler software was used for the enrichment analysis of differential genes.

#### 2.4.2. Quantitative PCR for mRNA Expression

Twelve genes were selected from the DEGs of each treatment for quantitative verification (Appendix A for primers). The reaction system was 15 µL, including 7.5 µL 2 × Power Green qPCR Mix, 0.3 µL upstream and downstream primers (10 µmol/L), 1.5 µL cDNA, and 5.4 µL ddH_2_O. Each sample was repeated three times and the reactions were run on a Roche LightCycler 96. The reaction procedure was as follows: 94 °C for 3 min; 94 °C for 15 s, 58 °C for 15 s, 72 °C for 20 s (fluorescence collection), 40 cycles. The specificity of the PCR product was detected by melting curve analysis. β-actin was used as the reference gene, and the expression level of the target gene was statistically analyzed using the 2^−ΔΔCt^ method.

### 2.5. Data Analysis

The results are presented as the means ± standard deviations. The data were tested for homogeneity of variance (visual inspection of residuals vs. fitted values), and the normality of the residuals was tested using the Shapiro–Wilk normality test. All of the response data of the corals were tested using a two-factor analysis of variance (ANOVA) with “temperature” and “algae” as the fixed factors, including the interaction term. Tukey’s test was used to identify significant differences between the temperature treatments. A post hoc Fisher’s least significant difference (LSD) test was used to determine the differences between the algal treatments. The data were analyzed and graphed using GraphPad Prism 8.0. *p* < 0.05 was considered a significant difference. Pictures were also drawn using Rstudio and chiplot (https://www.chiplot.online/ accessed on 6 November 2022).

## 3. Results

### 3.1. Results of Macroalgae on Physiological Processes

As shown in Figure 2A,B, at an ambient temperature, the LH corals experienced a 53.1% and 71.2% decline in the density and Chl *a* of zooxanthellae (*p* = 0.046, *p* < 0.01), respectively, than the LD corals. There was no significant discrepancy between the LH and LM treatments. With the rising temperature, the *A. hyacinthus* in the HD group decreased by 84.4% and 93.8%, respectively, compared with the LD group (*p* < 0.01), at which point the addition of *C. taxifolia* recovered the density and pigment content in the HH and HM groups. Hence, there was an interaction between the algae and the temperature treatment (*F* = 6.9, *p* = 0.02, *F* = 10.8, *p* < 0.01).

The constant high temperature and algae treatment elicited no response in terms of the protein and growth rates (Figure 2C,D). After 4 weeks of *A. hyacinthus* cocultured with *C. taxifolia* at 27 °C, the protein content of the tissue in the LH and LM nubbins reduced by 37% compared with the LD nubbins (*p* > 0.05). At 30 °C, the increased temperature had a visible 55% inhibitory impact on the tissue protein in the HD nubbins (*p* < 0.05), but did not provoke any distinct variations in either the HH or HM nubbins.

As indicated by the change in buoyant weight, the growth rate of the corals in the LD treatment group was the highest at 27 °C, with a mean value of 1.1 ± 0.6 mg cm^−2^ d^−1^. The coculture with macroalgae (whether in contact or not) decreased the growth rate of the coral by 31% (*p* > 0.05). During the upgraded temperature pressure, there was a slight inhibitory influence on the growth rate in the HD corals (*p* > 0.05). What is more, the elevating temperature resulted in the LM corals growing at the lowest rate, with a value of 0.5 ± 0.1 mg cm^−2^ d^−1^, but there was no difference among the treatments.

As shown in Figure 2E,F, the macroalgae treatments increased the antioxidant capacity of the corals under all conditions. At a surrounding temperature of 27 °C, the SOD activity increased from 207.3 ± 50.6 U mgprot^−1^ in the LH system to 371.5 ± 125.8 U mgprot^−1^ in the LD system, increasing to 393.3 ± 121.7 U mgprot^−1^ in the LM system (*p* > 0.05). Moreover, with the temperature climbing to 30 °C, the mean of the SOD in the HD system increased 2.3-fold compared to the LD system (*p* = 0.02). The levels in the LH and LM systems mildly rebounded from 40 to 48.1% compared to the HD system (*p* > 0.05), indicating that both factors did interact (*F* = 6.3, *p* = 0.02).

Simultaneously, at an ambient temperature, the activity of CAT increased from 8.4 ± 4.5 U mgprot^−1^ in the LD system to 14.8 ± 10 U mgprot^−1^ in the LH system, and continued to increase to 51.4 ± 2.6 U mgprot^−1^ in the LM system (*p* < 0.01). Among the treatments under rising temperature conditions, the activity of CAT in the HH system attained the highest levels at 45.2 ± 5.1 U mgprot^−1^ (*p* < 0.05) and the LM system was comparable to the LD system. However, the CAT responses of coral in higher temperatures were dramatic, and increased 3.1-fold in the HH system versus the LH system, and in the HM system was 58.5% less than in the LM system (*p* < 0.05). The combined effect of temperature and macroalgae affected the coral (*F* = 43.5, *p* < 0.01).

### 3.2. Results of Transcriptome Analysis of A. hyacinthus

After removing the sequencing adaptor and low-quality data, a total of 126,587,106 clean reads were obtained, and the Trinity mapping yielded 53,676 unigenes with an average assembly rate of 66.6%, of which 33 263 were longer than 1000 bp and 4011 bp were N50 in length. The SwissProt, COG, KOG, GO, KEGG, Pfam, and eggNOG databases were used to annotate the above unigenes and 14,559, 10,483, 18,073, 27,495, 22,743, 28,571, and 22,532 unigenes were annotated (Appendix A). Vividly depicted in Appendix B Figure A1 is a noteworthy scenario in which, in contrast to the HD group, the LD, HH, and HM groups converge to a single community, similar to the LH and LM groups.

As shown in Figure 3A,B, there were 12,034 and 12,473 DEGs in the LH and LM treatment groups compared with the LD group, among which several vesicle transport (Rab family), autophagy (ATG 5 and ATG10), and apoptosis (Casp7 and Casp8)-related genes were identified. This differs from the LD and HD treatments with 1601 DEGs, which were associated with calcium balance (CalM and Galaxin), oxidative stress (CAT, SODC and HSP family), and immune response (CYP3A, TLR2, TRAF3, MYD88, and IFI47). Respectively, the HH and HM treatments had 2877 and 328 DEGs. The DEGs in the HH treatment were related to stress immunity (SODC, HSP70, CYP3A, IFI47, and TRAF5) and apoptosis (FADD, HIRA2, Casp3, and Casp7) and the HM group exhibited partial immune-related DEGs (TNFRSF14 and GPX). The qPCR results of the 12 DEGs were consistent with the transcriptome analysis results, indicating the reliability of the transcriptome results of this experiment (Figure 3C,D).

The DEGs were conspicuously enriched in 257, 273, 175, 193, and 64 KEGG pathways in the LH, LM, HD, HH, and HM groups, respectively. Among them, *C. taxifolia* mainly affected FoxO, the mTOR signaling pathway, endocytosis, autophagy, apoptosis, etc., in the LH or LM groups (Figure 4a,b). The thermal stress principally affected the microbial immune response pathway of *A. hyacinthus*, such as the cytosolic DNA-sensing pathway, herpes simplex virus 1 infection, and the Toll-like receptor signaling pathway (Figure 4c). At high temperatures, the DEGs gathered in the NOD-like receptor and TNF cell immune signal transduction and apoptosis pathways when *C. taxifolia* was in indirect contact with *A. hyacinthus* (HH) (Figure 4d), while DEGs gathered in the immune pathway, such as in the cytokine–cytokine receptor interaction, when *C. taxifolia* was in direct contact with *A. hyacinthus* (Figure 4d).

## 4. Discussion

### 4.1. Effects of C. taxifolia on the Physiology and Ecology of A. hyacinthus

The symbiotic relationship between reef-building coral and zooxanthellae is an important foundation of coral reef ecosystems [30]. Zooxanthella play an irreplaceable role in promoting coral growth, reproduction, and immune response. Approximately 90% of corals’ energy supply comes from zooxanthella-photosynthesizing products, such as glycerol, glucose, and amino acids [31]. However, when environmental conditions shift dramatically, the coral might be unable to provide energy due to the accumulation of toxic oxygen free radicals or the damage of photosynthetic organs in zooxanthella, finally resulting in the excretion of the symbiotic algae [32,33,34]. In terms of the physiological performance in this study, it was found that, compared to the LD treatment, the density and Chl *a* content of zooxanthellae in *A. hyacinthus* in indirect contact with *C. taxifolia* (LH) decreased (*p* < 0.05), whereas both the SOD and CAT enzyme activities increased, but with no sign of coral bleaching. The adverse effects of macroalgae on symbiotic algae in coral have been widely discussed [10,11,12]. Rasher et al. [10] demonstrated that the coral genus *Porites* bleached after contact with the brown alga *Lobophora variegata* for 20 days, which could be related to the metabolites produced by macroalgae. Shearer et al. [35] found that the green alga *Chlorodesmis fastigiata* caused significant damage to the PSII of zooxanthella in *Acropora millepora* and *Montipora digitata*.

Moreover, the results were consistent with the transcription results, which showed that the genes associated with apoptosis (Casp3, Casp7, and HIRA2), endocytosis and autophagy (Rab10, Rab11A, and ATG5), and oxidative stress (CAT and Hsp40) were upregulated. Thus, the decrease in the density of zooxanthellae in the LH treatment may be regulated by the transport of coral vesicles. Corals obtain planktonic zooxanthellae from water through endocytosis and remove senescent and damaged zooxanthellae through endocytosis, autophagy, and apoptosis [36]. Firstly, PKC family proteins might be introduced into vesicle tube clusters through straightforward interaction with the tiny GTPase Rab2 and participate in vesicle transport, thereby facilitating autophagy [37]. Secondly, it is widely acknowledged that Rab GTP family proteins regulate vesicle trafficking and membrane fusion [38]. Rab5, as an upstream gene, regulates downstream Rab7 and 11. If Rab5 was absent, the antagonism of Rab7 and 11 would be triggered and the autophagy of zooxanthellae would be induced [39,40,41]. Thus, the decrease in the Rab5 expression might be connected to the promotion of the autophagy process accelerated by PRKCI. Furthermore, the ATG family is still overexpressed in autophagy. The ATG5 factor was deeply involved in several cellular processes, including the formation of autophagic vesicles, and might promote apoptosis when it is overexpressed [42]. Interestingly, the expression of pro-apoptosis genes, for example, Casp7, increased as well.

Here, the loss of zooxanthellae density and pigment content reflects the allelopathy impacts of macroalgae. Factually, it was confirmed that the adverse impacts of allelopathy on zooxanthellae might correlate with the dissolved organic carbon (DOC) and terpenoids that are released by macroalgae [11,43,44,45,46]. The DOC could promote bacterial metabolism on coral surfaces and competition for oxygen [45]. As a result, coral health is negatively affected. With the death of coral, more ecological niche on the coral reef is available for macroalgae to reproduce, forming a DDAM (DOC, disease, algae, and microorganisms)-positive feedback loop [46]. Terpenoids might also mediate changes in microbial community structure, resulting in the PSII system injury of zooxanthellae [11].

### 4.2. Effects of Thermal Stress on the Physiology and Ecology of A. hyacinthus

In this study, the high temperature (HD) triggered *A. hyacinthus* bleaching and the activities of SOD and CAT enzymes simultaneously enhanced, a phenomenon that has already been confirmed [47,48,49]. Thermal stress can cause the host coral or zooxanthellae algae to produce large amounts of superoxide anion radicals (ROS), as well as an increase in pathogenic microorganisms that poison coral cells [50,51,52,53]. In this process, host cells may regulate the corresponding proteins to resist thermal stress through gene expression; this was also found in the gene expression analyses of this study.

The results showed that the CalM expression was inspired by thermal stress. This could induce the release of Ca^2+^ from the endoplasmic reticulum (ER) storage into the cytoplasmic matrix and activate biological processes such as oxidative stress and the immune response [54]. In terms of oxidative stress, several factors involved in the scavenging of free radicals, anti-damage, and immunity enhancement were identified, including CAT, SODC, and the Hsp family [55]. Commonly, ROS would be scavenged by SOD and converted to H_2_O_2_ during oxidative metabolism, which requires CAT to decompose H_2_O_2_ in order to prevent peroxidation. Furthermore, it was previously noted that warning indicators of coral bleaching, such as heat shock proteins and molecular chaperones, function in a variety of cellular processes to maintain physiological functions such as protein folding, intracellular protein transport, and resistance to protein denaturation [56].

At the same time, the transcription results showed that the host was resistant to the invasion of pathogenic microorganisms through gene expression. Obviously, the function of the Toll-like receptor signaling pathway could mediate antifungal infection semaphore transduction [57]. In this pathway, the TLR2 and TRAF3 factors were upregulated. The TLR2 was involved in pathogen recognition and TRAF3 was associated with CD40 signal transduction, all of which had the function of activating an immune response [58,59]. Furthermore, this study discovered that the TRAF family regulates the downstream expression of IFI47 in the TNF signaling pathway by binding to TNF receptors, and the protein encoded by IFI47 might trigger cell resistance to viral and microbial infections. Besides those, the CYP450 superfamily was also upregulated among the immune genes, which could catalyze the oxidation of various organic substances, adjust microorganisms, and play a dominant role in detoxification [60,61].

Hence, the factors Apaf, FADD, Casp3, and Casp7 gained an increasingly upward momentum and ultimately promoted apoptosis, because of an adequate ROS elimination or immunological response to pathogenic bacteria under rising temperatures [53,62].

### 4.3. Combined Effects of C. taxifolia and Thermal Stress

Physiologically, one pattern that emerged in this study was that the bleaching rate due to increasing temperature was approximately mitigated or even eliminated when *A. hyacinthus* interacted with *C. taxifolia*. In addition, different from the HD treatments, the SOD enzymes in the HH group maintained a shrinking tendency, while the CAT enzymes increased. The comparison between the HM and HD groups showed that the SOD and CAT enzymes diminished simultaneously, which might be due to the damage of the coral tissue owing to the physical friction of *C. taxifolia*. Some studies have reported that macroalgae could promote the physiological functions of coral under thermal stress. Brown et al. [62] found that the effect of contact with the macroalgae *Halimeda heteromorpha* led to an obvious increase in the photosynthesis-to-respiration ratio (P_netmax_/R_dark_) relative to the coral *Acropora intermedia* at 30 °C. The presence of algae might intermittently provide shade, which can ensure free flow, keeping the exchange of abundant oxygen due to the coarsening stolon and avoiding the thermal process because of the foliaceous [63,64]. However, the degradation of oxidative stress kinase has scarcely been observed.

The transcriptome results showed that the oxidative stress-related genes such as the SODC and HSP family were downregulated when the coral was co-cultured with macroalgae at 30 °C. In addition, in contrast to the HD treatment, the DEGs in the HH group were predominantly enriched in the NOD-like receptor signaling pathway, which was associated with pathogen recognition and immune response [65]. Immediately afterward, the TRAF family and MYD88 genes in this pathway, through downregulation, regulated several pathways including the Toll-like receptor signaling pathway and TNF signaling pathway, which jointly attenuated the immune stress produced by microorganisms. Yet, the apoptosis pathway also experienced remission, with the HIRA2, Casp3, and Casp7 factors markedly on the decline. The HM treatment principally acted on cytokine–cytokine receptor interaction and glutathione metabolism distinct from the HD group, which was helpful for promoting immune activity.

It has been thought-provokingly reported that the fade in antioxidant enzymes under warming conditions might be relevant to DMSP (dimethyl thipropionate) released by zooxanthellae and macroalgae [66]. Raina et al. and Hopkins et al. [67,68,69] asserted that DMSP could be obtained from the outside or synthesized by *Acropora*, which were translated into DMS (dimethyl sulfur) in the process of scavenging ROS and the digestion of oxidative stress during environmental stress. Therefore, the DMSP secreted by algae might be one of the reasons for which *C. taxifolia* could alleviate the thermal stress of the coral *A. hyacinthus*.

## 5. Conclusions

This study explored the crucial issue of how the physiology and molecular response of the hermatypic coral *A. hyacinthus* was affected by the foliaceous macroalgae *C. taxifolia* at various temperatures. Therefore, the different ways including indirect and direct co-culturing with seaweed were set up at an ambient temperature (27 °C) and at a +3 °C increase in the temperature. Firstly, the results demonstrated that macroalgae could prominently trigger a drop in the density of zooxanthellae at various temperatures, which has been associated with the biological processes of vesicle transport, autophagy, and apoptosis regulated by the Rab5, ATG5, and Casp7 transcription factors. Secondly, oxidative stress (CAT, SODC, and the HPS family) and microbial immune response (IFI47 and the TRAF family) biological processes were violently aggravated by heat stress, resulting in cell apoptosis, at which point *C. taxifolia* alleviated the pressure influences.

## Figures and Tables

**Figure 1 biology-11-01792-f001:**
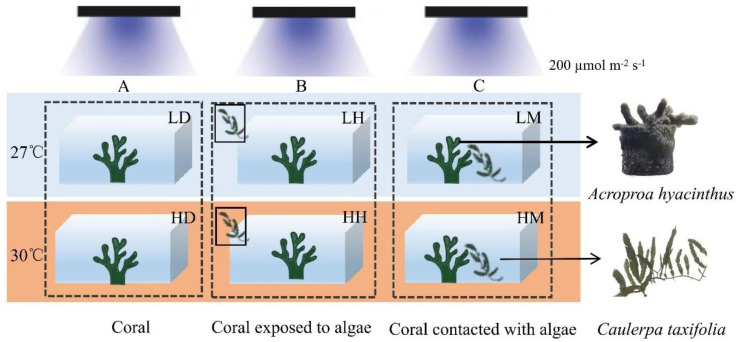
Experimental design. (**A**) Control group, (**B**) indirect contact group, (**C**) direct contact group. The orange and blue colors represent the different temperature treatments (27 °C and 30 °C). The conditions for the treatments are presented in Appendix A. Three aquariums were set for each treatment, *n* = 3.

**Figure 2 biology-11-01792-f002:**
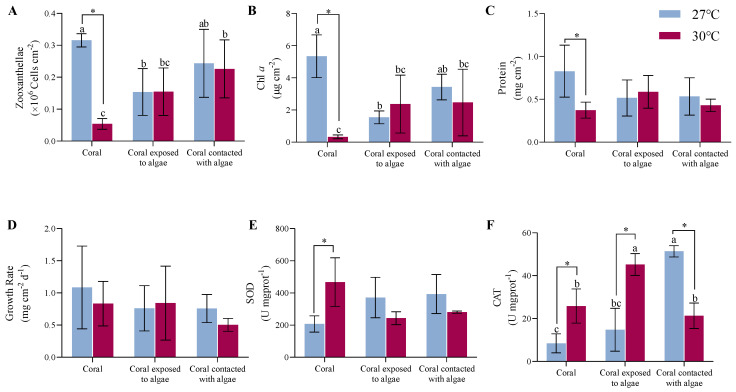
Summary of physiological indicators. The effect of different temperatures of 27 °C (blue) and 30 °C (red) on the (**A**) zooxanthellae, (**B**) Chl *a*, (**C**) protein content, (**D**) growth rate, (**E**) SOD, and (**F**) CAT of macroalgae-treated corals after 4 weeks of the experiment. Different letters indicate that there are significant differences between the macroalgae treatments at the same temperature (*p* < 0.05), * indicates that there are significant differences between the different temperatures (*p* < 0.05). Data are expressed in terms of the mean ± standard deviation, *n* = 3. The source data and two-factor analysis of variance (ANOVA) data are provided as a source data file in Appendix A.

**Figure 3 biology-11-01792-f003:**
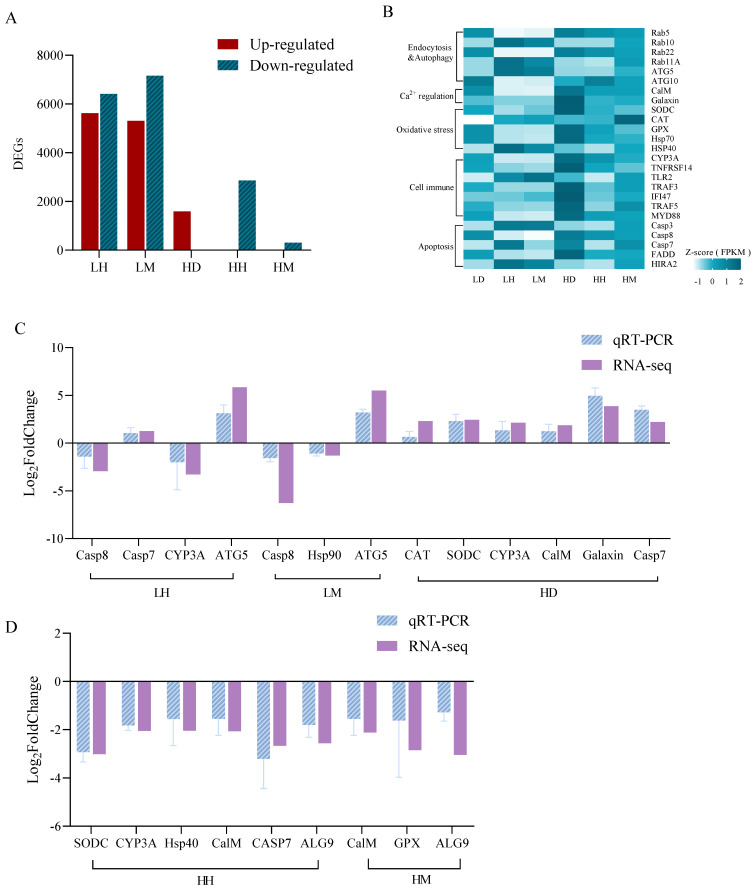
Gene expression dynamics of *A. hyacinthus* after 4 weeks of the experiment. (**A**) DEGs statistics in all treatments. Select FDR < 0.001 and shown as LH, LM, and HD groups vs. LD group, HH and HM groups vs. HD group. (**B**) FPKM expression map of different biological processes regulated by DEGs. (**C**) Compared with the LD group, the DEGs verification of LH, LM, and HD groups. (**D**) Compared with the HD group, the DEGs verification of HH and HM. The source data are provided as a source data file in Appendix A.

**Figure 4 biology-11-01792-f004:**
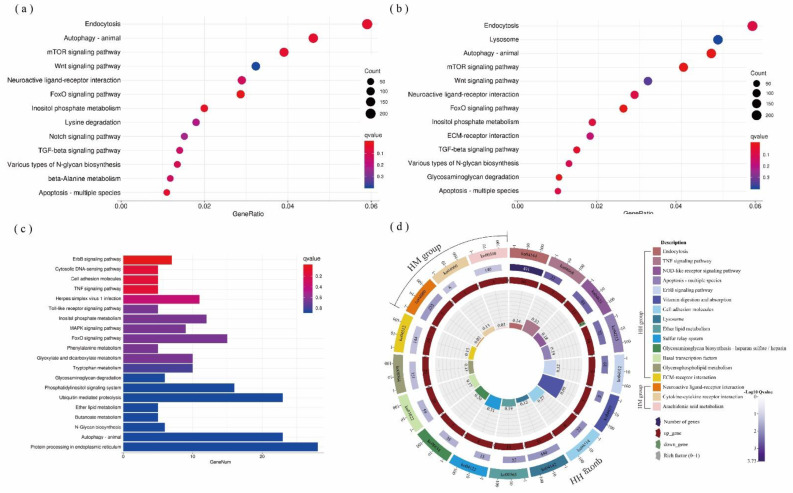
Molecular plasticity explains the effects of macroalgae on corals. Distribution of gene ontologies which were significantly enriched in the KEGG pathways of (**a**) LH, (**b**) LM, and (**c**) HD treatments. (**d**) HH and HM treatments’ KEGG enrichment pathway (*p* < 0.05). Colors represent KEGG pathway description and genes description (up or down). Rich factor represents DEGs divided by total genes in this pathway. The source data are provided as a source data file in Appendix A.

## Data Availability

The data and calculation tools are available from the corresponding author upon reasonable request (zjouliuli@163.com).

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
