# Peer review of "Effects of Caulerpa taxifolia on Physiological Processes and Gene Expression of Acropora hyacinthus during Thermal Stress"

_biology, 2022, doi:10.3390/biology11121792_

Round 1

Reviewer 1 Report

Dear Editor, 

In the present study, the authors try to study the effects of Caulerpa taxifolia on physiological processes and gene expression of Acroproa hyacinthus during thermal stress. In general, the text is organized well. The results are well presented. But some minor corrections are needed. 

Author Response

Dear reviewer, thank you for your suggestions on my article. The reply to your questions is attached in the form of world, please check it.

Author Response

(The authors gave the same response as above.)

Reviewer 3 Report

Dear Author,

I am suggesting some suggestions regarding your original research work paper. Please notice this suggestions. I have attached the world file.

With Regards

Author Response

(The authors gave the same response as above.)
